# Planning By Active Sensing

**Kaushik Lakshminarasimhan**                                                    JL5649@COLUMBIA.EDU
*Zuckerman Mind Brain Behavior Institute, Columbia University*

**Seren Zhu**                                                                     LT1686@NYU.EDU
**Dora Angelaki**                                                                 DA93@NYU.EDU
*Center for Neural Science, New York University*

## Abstract

Flexible behavior requires rapid planning, but planning requires a good internal model of the environment. Learning this model by trial-and-error is impractical when acting in complex environments. How do humans plan action sequences efficiently when there is uncertainty about model components? To address this, we asked human participants to navigate complex mazes in virtual reality. We found that the paths taken to gather rewards were close to optimal even though participants had no prior knowledge of these environments. Based on the sequential eye movement patterns observed when participants mentally compute a path before navigating, we develop an algorithm that is capable of rapidly planning under uncertainty by active sensing i.e., visually sampling information about the structure of the environment. New eye movements are chosen in an iterative manner by following the gradient of a dynamic value map which is updated based on the previous eye movement, until the planning process reaches convergence. In addition to bearing hallmarks of human navigational planning, the proposed algorithm is sample-efficient such that the number of visual samples needed for planning scales linearly with the path length regardless of the size of the state space.

**Keywords:** model-based reinforcement learning, sequential decision-making, navigation, maze, gaze

## 1. Introduction

Planning, or the ability to flexibly choose a sequence of actions in a goal-dependent manner, is a cornerstone of human intelligence. Signatures of human planning have been documented in a variety of sequential decision-making paradigms ranging from simple two-step decision-making tasks (Daw et al., 2011; Miller et al., 2017; da Silva and Hare, 2020) to more complex, multi-step navigation tasks (Simon and Daw, 2011; Anggraini et al., 2018; Zhu et al., 2022; de Cothi et al., 2022). A wealth of data suggests that human planning exhibits two key properties – *computational efficiency* i.e., the convergence time of the planning algorithm should not scale rapidly with the size of the state space (Kool et al., 2017), and *noise robustness* i.e., the algorithm should overcome any uncertainty in the representation about the model of the world (Hudson et al., 2008; Alhussein and Smith, 2021).

Early models of planning in artificial intelligence systems were formulated as heuristic forward search algorithms (Hart et al., 1968; Pearl, 1984; Korf, 1985) or their backward counterparts (LaValle, 2006) that operated on symbolic world models. The use of appropriate heuristics allows for focusing computations on the task-relevant parts of the state

space, making these algorithms computationally efficient. However, since these models express transition dynamics in symbolic terms, they are not inherently capable of dealing with subjective uncertainty in the transition dynamics.

An alternative formulation of the planning problem using the framework of Markov Decision Process (MDP) has emerged as a standard approach to deal with stochastic transition dynamics (Sutton and Barto, 2017). Planning in the MDP framework is typically solved via dynamic programming (DP) algorithms such as value iteration or policy iteration (Bellman, 1957; Howard, 1960) which entail alternating between policy evaluation and policy improvement. Unlike heuristic search, DP algorithms converge to the optimal solution even if the transition dynamics are stochastic. Model-based planning algorithms such as Dyna (Sutton, 1991; Moerland et al., 2020) allow for reducing stochasticity due to subjective uncertainty about the transition dynamics (epistemic uncertainty) by performing model updates in conjunction with policy updates. However, a common drawback of DP algorithms is that the policy updates are performed across the entire state space in an undirected manner, making them much less computationally efficient than heuristic search.

## 2. Related work and our contribution

Studies on planning have shown that humans minimize computational complexity using strategies such as pruning and decomposition of decision trees (Huys et al., 2012; Solway et al., 2014; Huys et al., 2015), resource-rational planning (Callaway et al., 2018; Ho et al., 2020), or by constructing simplified mental representations of the world (Ho et al., 2022). While such insights help identify useful model classes, they are insufficient to construct a granular algorithm of human planning at the level of individual planning steps. Moreover, above studies do not account for representational noise and thus do not explain how we might plan when our model of the environment is imprecise or wrong. Humans rely on structured eye movements to reduce uncertainty in simple discrimination/detection tasks (Renninger et al., 2007; Yang et al., 2016; Hoppe and Rothkopf, 2019) but it is not yet clear if and how such strategies are used when planning action sequences in complex environments. Recently, recurrent neural network models have been used to explain human eye movements during memory-guided (Lakshminarasimhan et al., 2018; Stavropoulos et al., 2023) and maze-solving (Li et al., 2023; Kadner et al., 2023) paradigms. Such models help identify the computational objective driving eye movements but do not explain the mechanisms by which individual eye movements are chosen in real time.

To develop a granular, algorithmic theory of human planning that accommodates a role for active sensing, here we first analyzed data gathered from a free-form behavioral experiment in which human participants used a joystick to navigate mazes in virtual reality. We found that nearly all trials comprised of an initial planning phase during which participants visually explored a small set of relevant states before navigating to the goal suggesting that planning was performed ahead of time via active sensing rather than in real time. Furthermore, visual exploration of those states unfolded in a sequential manner until participants were able to successfully connect the start and goal states. Based on this spatiotemporal pattern of visual sampling, we propose an algorithm for planning under uncertainty by active sensing whereby an internal model of the world is updated in conjunction with planning

steps in an incremental fashion by visually sampling new states along the local gradient of the subjective value landscape.

In the following sections, we provide the mathematical foundations of navigational planning in the MDP framework, describe key results of the human behavioral experiments, describe the planning algorithm, evaluate its performance, and analyze the sample efficiency.

## 3. Preliminaries

Navigation can be formulated as a Markov decision process (MDP) described by the tuple $\mathcal{M} = \langle \mathcal{S}, \mathcal{A}, P, R, s_0 \rangle$ whose elements denote, respectively, a finite state space $\mathcal{S}$, a finite action space $\mathcal{A}$, a state transition distribution $P(s'|s, a)$, a reward function $R(s)$, and an initial state $s_0 \in \mathcal{S}$. Given that an agent is in state $s \in \mathcal{S}$, the agent may execute an action $a \in \mathcal{A}$ in order to bring about a change in state from $s$ to $s'$ with probability $P(s'|s, a)$ and harvest a reward $R(s)$. In the case that an agent is tasked with navigating to a goal state $s_G$ where the agent would receive a reward, the reward function $R(s) = \delta(s - s_G) - 1$ such that the reward is concentrated in the goal state. Given this formulation, we may compute the optimal policy $\pi_*(a|s)$, which describes the actions that an agent should take from each state in order to reach the goal state in the fewest possible number of steps. The optimal policy may be derived by computing optimal state values $V_*(s)$, defined as the expected future rewards to be earned when an agent begins in state $s$ and acts in accordance with the policy $\pi_*$. The optimal value function must satisfy the Bellman optimality equation:

$$V_*(s) = \max_a \sum_{s'} P(s'|s, a)[R(s) + V_*(s')] \tag{1}$$

The optimal policy is given by the argument $a$ that maximizes the right-hand side of Equation 1. Intuitively, the optimal policy corresponds to ascending the value function $V^*(s)$ where the value gradient is most steep. The standard approach to solve the Bellman optimality equation is through dynamic programming (DP) algorithms such as value iteration or policy iteration. These algorithms proceed by choosing a random policy $\pi(s|a)$ and repeatedly applying elements of the above equation – policy evaluation and policy improvement – in alternation until convergence. Since each application propagates value between neighboring states, this operation is typically referred to as a Bellman backup or just 'Backup'. The policy evaluation step computes the state-value function $V_\pi(s)$ when acting according to policy $\pi$:

$$V_\pi(s) = \sum_a \pi(a|s) \sum_{s'} P(s'|s, a)[R(s) + V_\pi(s')] \tag{2}$$

The solution to Equation 2, given by $V_\pi(s) = (I - P\pi)^{-1} R(s)$, is the value function used for the policy improvement step in policy iteration. In the policy improvement step, a new policy $\pi'(a|s) = \delta(a - a_*(s))$ is chosen such that:

$$a_*(s) = \arg\max_a \sum_{s'} P(s'|s, a)[R(s) + V_\pi(s')] \tag{3}$$

The DP algorithm converges when $\pi' = \pi = \pi_*$ or equivalently when $V_\pi = V_{\pi'} = V_*$.

## 4. Results

### 4.1. Behavioral task

To model human planning, we used data from a virtual reality (VR) task in which participants navigated to cued reward locations in hexagonal mazes. The experiments used a head-mounted VR system (HTC Vive Pro) with a wide field of view to provide an immersive experience. Participants ($n = 13$) viewed the environment from a first-person perspective and freely rotated in a swivel chair and used an analog joystick to control their forward and backward motion along the direction in which they were facing (Figure 1A). The program recorded their position in the virtual maze as well as their gaze using a built-in eye tracker. To facilitate quantitative analyses, mazes were designed with a hidden underlying triangular tessellation where each triangular unit constituted a *state* in a discrete state space (Supplementary Figure 1A) but this discretization was invisible to participants. A fraction of the edges of the tessellation was chosen to be impassable barriers (obstacles). Participants could take *actions* using the joystick to achieve *transitions* between adjacent states which were not separated by obstacles. Critically, participants experienced a relatively high vantage point and were able to gaze over the tops of all of the obstacles to gather information about distal transitions through visual exploration (Figure 1A). On each trial, participants had to collect a reward by navigating to a random reward location drawn uniformly from all states in the maze. They had to locate the reward (a banana) and navigate to it after which a new reward for the next trial was spawned without breaking the continuity of the task. In separate blocks, participants navigated to 50 goals in each of five different mazes of variable complexity (Supplementary methods). The simplest of these was an open maze that required no planning, while the rest were structured.

### 4.2. Signatures of planning

We found that participants navigated to the goal along optimal trajectories in all mazes even though they had no prior experience in any of them i.e., they did not have a model of the environment (Figure 1B; summarized in Supplementary Figure 1B). This suggests that they *planned* their trajectories before navigating, as confirmed by their velocity traces which showed that participants were typically stationary for a brief period at the beginning of each trial ('planning period') and then navigated without stopping (Figure 1C). Planning duration increased with navigation duration in structured (but not open) mazes implying that planning in these mazes was effortful and that the complexity of planning increased with the path length (Figure 1D).

### 4.3. Active sensing strategy

Since participants did not have a model of the environment at the beginning of the trial, we reasoned that they built a model by visually exploring the maze during the planning period i.e., by *active sensing*. Therefore, the source of algorithmic complexity is twofold: sample complexity associated with the number of visual samples needed to build a sufficiently good model, and computational complexity associated with performing Bellman backups to infer a good policy. Since random visual exploration is sample inefficient, we hypothesized that participant's active sensing strategy could be dictated by task demands. To test this, we

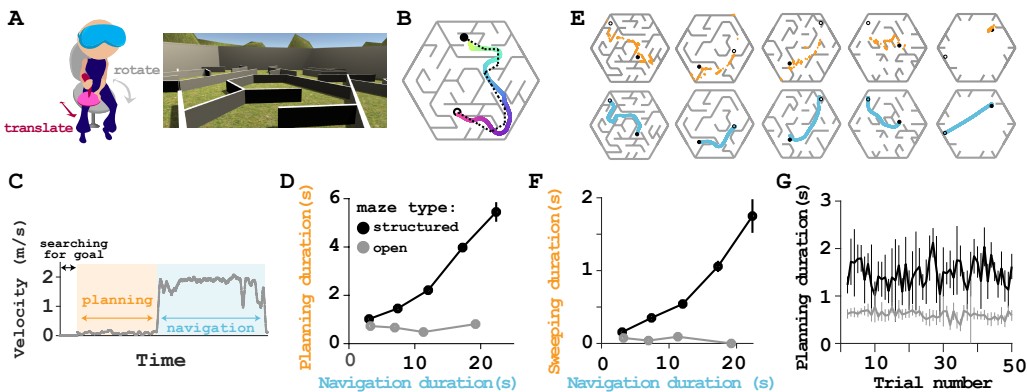

Figure 1: **Human behavior. A**. Humans navigated in unfamiliar mazes using a joystick. **B**. An example trial: navigated trajectory (color) with the optimal trajectory overlaid (dashed). Open and solid circles denote starting location and reward location respectively. **C**. Trials typically comprised an initial planning period followed by a navigation period. **D**. Mean planning duration as a function of mean navigation duration across trials. **E**. Example trials showing gaze positions during planning (yellow dots) and the trajectories navigated subsequently (blue traces). **F**. Mean sweeping duration as a function of mean navigation duration. **G**. Mean planning duration as a function of experience.

analyzed their gaze positions during the planning period relative to the trajectory taken while navigating and found two salient features. In the spatial domain, there was a striking correspondence between the two in structured (but not open) mazes suggesting that the active sensing strategy was trial-specific (Figure 1E). Note that this correspondence was not perfect since the trajectories are determined only after the planning process is complete. In the temporal domain, participants performed visual sweeps during which they sampled the states that comprised the trajectory in a sequential manner (Supplementary Figure 2). The total duration of these sweeps increased with the navigation duration since more states need to be traversed as trajectories get longer (Figure 1F). Finally, we note that the planning duration was stable across trials within a block (Figure 1G) suggesting that information gathered about the model was not consolidate across trials but rather learned from scratch on each trial. This is most likely because even the least complex among the structured mazes had a fairly complex transition structure and the points of view differed across trials.

Taken together, the spatiotemporal pattern of gaze during planning suggests that the sampling strategy was: (i) influenced by the starting and reward locations, and (ii) informed by a heuristic which encouraged the exploration to evolve sequentially. In the next section, we propose an algorithm for planning by active sensing that incorporates this sparse sampling strategy to update the world model. However, we do not exactly know how humans reduce the computational complexity i.e., how are their backups organized? We make a parsimonious assumption that Bellman backups are performed only for the set of sam-

pled states and immediately after each model update. We demonstrate that this minimal assumption is sufficient to generate a successful plan.

## 4.4. Planning algorithm

Based on the observations from human behavior, we developed a planning algorithm whose core components are described below and illustrated in Figure 2.

**Model uncertainty.** Without loss of generality, we consider navigation in deterministic mazes such as the ones used in the human experiments. For these mazes, the transition dynamics $P_{\text{true}}(s'|s, a) = 1$ if an action $a$ allows a transition from state $s$ to a neighboring state $s'$ and $P_{\text{true}}(s'|s, a) = 0$ otherwise. In case an action fails to bring about a change of state due to the presence of an obstacle, then $P_{\text{true}}(s'|s, a) = \delta(s' - s)$. Note that if the agent has a perfect model of the environment, the subjective transition dynamics upon which the planning algorithm operates would be identical to these deterministic transition dynamics. In the other extreme scenario where the agent has no knowledge of the model, $P_{\text{noise}}(s'|s, a) = P_{\text{noise}}(s|s, a) = \frac{1}{2}$, corresponding to equal probabilities of success and failure of an action to bring about a change of state from $s$ to $s'$. In practice, the subjective model might be somewhere in-between and we capture this by assuming that the subjective transition model is a weighted sum of the true and noisy transition models:

$$P(s'|s, a) = (1 - \alpha)P_{\text{true}}(s'|s, a) + \alpha P_{\text{noise}}(s'|s, a) \tag{4}$$

where $\alpha \in [0, 1]$ denotes the level of uncertainty.

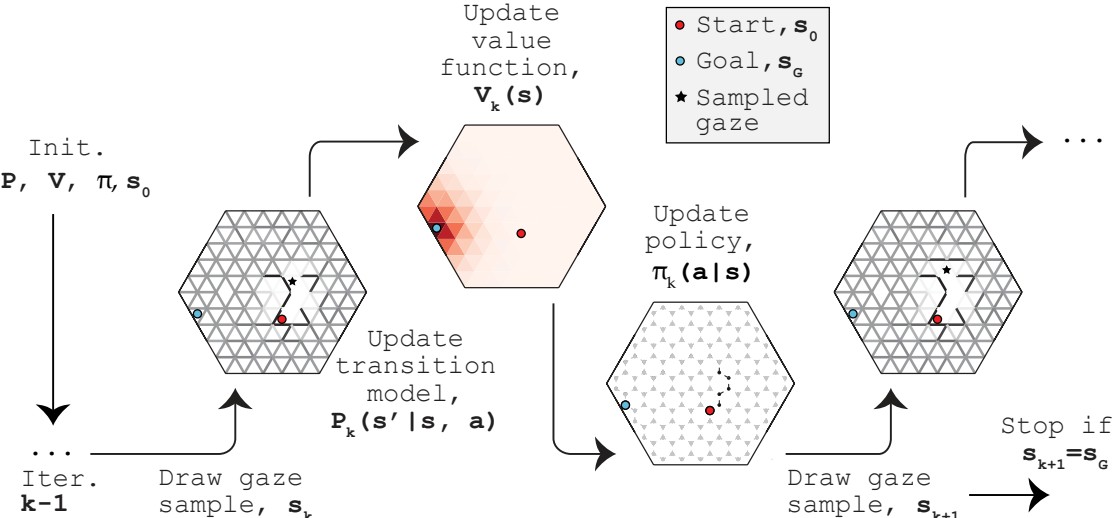

Figure 2: **Planning algorithm.** Each iteration of the algorithm updates the transition model locally around the state sampled by the gaze. The updated transition model is used to update the value function based on the current policy before performing a policy update.

**'Online' update.** In standard DP algorithms, policy improvements (equation (3)) are performed for all states $s \in \mathcal{S}$. This ignores knowledge of the initial state $s_0$ making them undirected and thus computationally inefficient. In contrast, adaptive real-time dynamic programming (RTDP) algorithm and its variants restrict model/policy updates to a single state and proceed by acting according to the current best policy at that state i.e., they interleave steps of planning and action selection. For this reason, they are considered as online planning strategies. We adopt a similar approach in our algorithm albeit with a subtle but important difference. Like human participants, we endow our agents with active sensing to learn about the consequences of actions at distal states by visually sampling them without physically navigating to those states. Although the resulting model/policy update equations at the sampled states are mathematically equivalent to online algorithms that perform the selected actions, we interpret them as simulated actions rather than real actions. This allows the agent to stay put at $s_0$ throughout planning. This difference can be formally expressed by defining a simulated MDP $\widetilde{\mathcal{M}}$ that is identical to the ground MDP $\mathcal{M}$, running the algorithm on $\widetilde{\mathcal{M}}$ to determine a policy $\widetilde{\pi}$, and then finally setting $\pi = \widetilde{\pi}$ before applying it on the ground. We skip this formalism to keep notations simple.

**Greedy sampling for model and policy updates.** Each iteration includes a model update step in which the subjective transition matrix is updated locally around the chosen state. Model updates are followed by policy evaluation, and policy update in the chosen state. Updates are performed in the initial state $s_0$ in the first iteration, and an action $a$ is simulated according to the best action for that state according to the current policy. This determines a new sample state $s_1$ for updating in the next iteration, following which the best action is simulated in that state to determine $s_2$ and so on until the $k^{\text{th}}$ sample $s_k = s_G$. Since new samples are generated by simulating actions at the currently sampled state, sampling takes place sequentially similar to human participants. However, since greedy sampling corresponds to ascending the value gradient, the sampled sequence may be sensitive to the assumptions we make about the value function before the transition model is learned. We demonstrate that a simple initialization strategy that depends only on the knowledge of the reward location works very well in practice (see below).

**Model update.** The transition model update in the $k^{\text{th}}$ iteration involves modifying the subjective transition dynamics $P_{k-1}$ according to:

$$P_k(s'|s,a) \leftarrow P_{k-1}(s'|s,a) + W_k(s)\Delta P_{k-1}(s'|s,a) \tag{5}$$

where $\Delta P_{k-1}(s'|s,a) = P_{\text{true}}(s'|s,a) - P_{k-1}(s'|s,a)$ and $W_k(s) = e^{\frac{-d(s,s_k)^2}{2\sigma^2}}$ is a Gaussian weight profile that decays as a function of distance $d(s,s_k)$ from the state $s_k$ sampled in the $k^{\text{th}}$ iteration. This weight profile was chosen to restrict information gathering to the immediate vicinity of the sampled state and is meant to mimic the effects of filtering by the human fovea.

**Initialization.** The initial sample is assumed to be at the starting location $s_0$. The initial transition model $P_0$ is given by equation (4) where we assumed $\alpha = 1$ which corresponds to maximum uncertainty. The policy is initialized to be a random walk such that

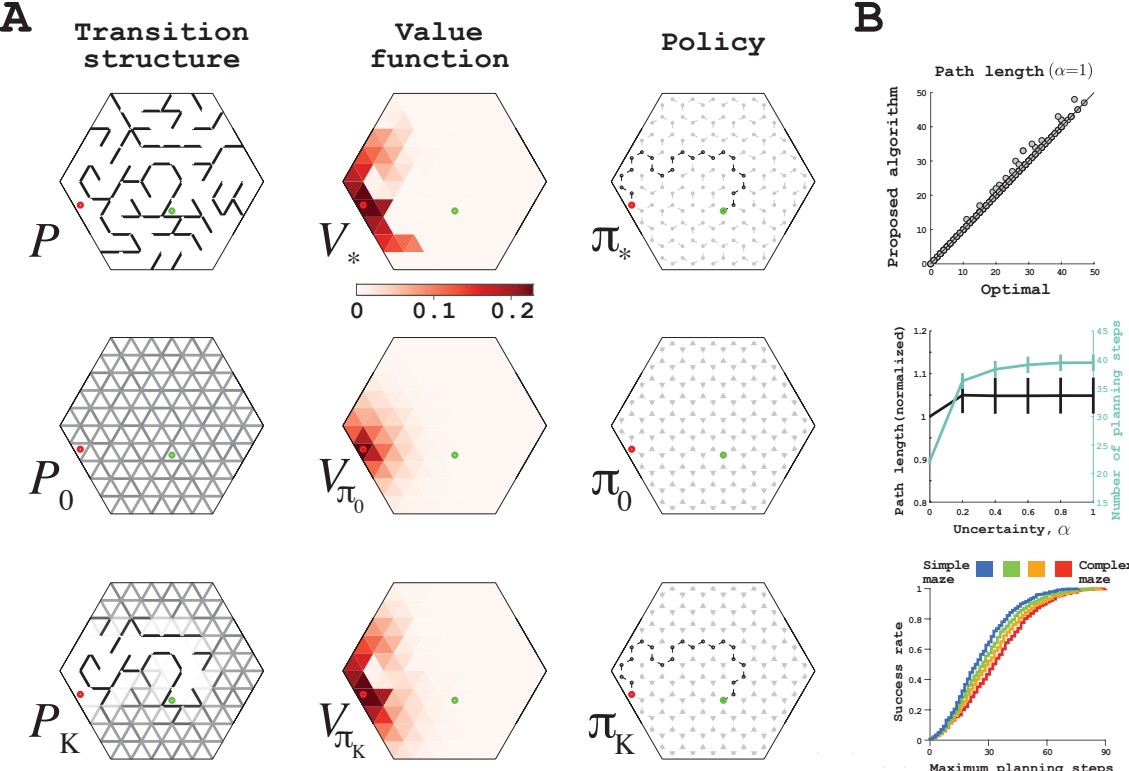

Figure 3: **Model performance. A.** Example simulation. *Top row*: The true transition
dynamics (*left*), and the corresponding optimal value function (*middle*) and opti-
mal policy (*right*) computed using the standard dynamic programming approach
for an example trial. Starting location and goal location are shown in green and
red respectively. Policy unfolding from the starting location is highlighted in
black. *Middle row*: The assumed transition dynamics, value function and (ran-
dom walk) policy at initialization ($k = 0$). *Bottom row*: Similar to middle panels,
but after convergence of the proposed algorithm ($k = \mathrm{K}$). A video of the full sim-
ulation is available at https://tinyurl.com/planning-algo. **B.** *Top*: Comparison
of the path length of the policy determined by the algorithm against the opti-
mal path length across trials *Middle*: Path length (normalized by optimal, black)
and number of planning steps (cyan) as a function of initial model uncertainty.
*Bottom*: Success rate as a function of the number of planning steps.

$\pi_0(a|s) = 1/M \ \forall s \in \mathcal{S}$ where $M = 3$ is the number of actions available in each state due
to triangular tessellation of the state space. These choices imply an initial value function
$V_{\pi_0}(s) = (I - P_0 \pi_0)^{-1} R(s)$ where the reward function $R(s) = \delta(s - s_G)$ is the only term
that is trial-specific.

To understand how the algorithm plans a trajectory, consider the example task shown
in Figure 3A where the objective is to determine the optimal policy to navigate between the
states marked by the colored circles (from green to red). The panels in the top row show the

mathematical quantities determined by a standard dynamic programming (DP) approach (policy iteration). This technique uses the true transition model (configuration of the maze, top row – left) to calculate the optimal state value function (heat map, top row – middle) and the corresponding optimal policy (vector field, top row – right). The value function and policy determined by DP serves as the ground truth for assessing the performance of the proposed algorithm. In contrast to DP techniques, we assume that the subjective transition model to be noisy such that the resulting model is very fuzzy with transitions between all neighbors having a probability close to 0.5 (middle row – left). Under this noisy transition model, a naive random walk policy assumed at initialization (middle row – right) evaluates to a value function that is markedly different from optimal (middle row – middle panel) before any updates are applied. Despite this conservative initialization, the algorithm converges to the true optimal policy (bottom row – right) unfolding from the start towards the goal and the corresponding value function resembles the optimal value function (bottom row – middle). In contrast, the subjective transition structure is still noisy in large swathes of the state space outside of the regions sampled by the algorithm (bottom row – left). Thus, the algorithm is able to determine the optimal policy with only a modest number of samples. A closer inspection of the policy function determined by this algorithm shows that, unlike the policy determined by dynamic programming, this policy is essentially random everywhere except for the sampled states. This demonstrates the directed nature of this algorithm imposed by the greedy sampling such that the final policy is only applicable to a subset of the state space. The specific subset depends on the subjective transition model, the starting state, and the goal state. Nevertheless, this specificity is precisely what makes the planning algorithm achieve low sample and computational complexity. An animation showing how the plan is gradually built up is available at https://tinyurl.com/planning-algo. Figure 3B summarizes the performance of the algorithm. The algorithm converges to the optimal solution on most trials (top) even when the model is completely unknown ($\alpha = 1$, middle) and across mazes of varying complexities (bottom).

The sampling strategy of the algorithm can be better understood by examining the spatiotemporal dynamics of the samples $s_k$ (Figure 4A) alongside the dynamics of the state value function, $V(s_k)$, of the sampled states (Figure 4B). Since the algorithm follows a greedy sampling strategy that follows the spatial gradient of the value function, the value of the sampled states generally increases. However, there is a precipitous drop in the middle of planning after about $k = 17$ iterations in example 1 (left panel, black triangle). Why does this happen? Notice that the new sample encountered on this iteration of planning reveals a (previously unknown) obstacle preventing access towards the goal state. When this new information about the transition model is used to perform model update (Equation 5), it results in downgrading of the values of the states affected by that obstacle during the policy evaluation step (Equation 2) which triggers a visual detour in the sampling strategy after the policy improvement step (Equation 3). On the other hand, value increases monotonically when samples are relatively unsurprising as in example 3 (right panel). Across simulations, we found that non-monotonic value dynamics are more prevalent in the most complex arena (65% of the trials) compared to the simplest one (12%). These findings illustrate how planning in the real-world can benefit from a rich interplay between information gathering and incremental Bellman backups. Importantly, the subjective value dynamics accompanying visual exploration can serve as a prediction for neuroscience experiments

that probe the neural basis of visually-guided navigation at a single-trial resolution (Gulli et al., 2020; Noel et al., 2022; Lakshminarasimhan et al., 2023). Specifically, we predict that activity dynamics of neurons encoding the subjective value should follow a temporal profile shown in Figure 4B when the spatiotemporal profile of gaze is given by Figure 4A.

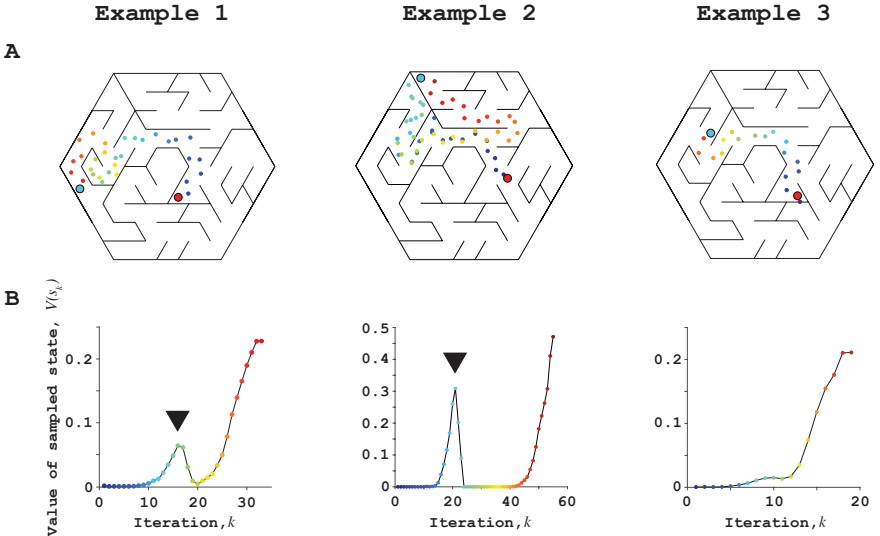

Figure 4: **Interplay between information gathering and value updating. A**. Spatial locations (jittered to avoid overlap) sampled by the proposed algorithm during three example trials. **B**. Subjective value of the sampled states, $s_k$ as a function of planning iteration, $k$ on the same trials. The example simulated in Figure 3 is shown in the leftmost panels. Note the decrease in value during the course of planning in spite of following a greedy sampling strategy in two of these examples. This decrease happens when newly gathered information unexpectedly reveals the presence of a previously unknown obstacle on the path towards the goal. Large red and cyan circles denote starting location and goal location. Gaze samples are color coded to denote the sequence of planning iterations (blue to red).

## 4.5. Model comparison

To test whether the particular strategy of sampling information in a sequential manner is in fact efficient, we constructed a variant of the algorithm in which the sampling strategy corresponded to random exploration. The policy updates still happened in a sequential manner in this variant. Across simulations of trials across mazes used to test human participants, we found that the median number of backups (planning steps) needed for convergence was indeed substantially reduced when sampling was performed sequentially in a coordinated manner with the policy updates (Figure 5A; sequential sampling: $16 \pm 2$ planning steps, random sampling: $57 \pm 6$ planning steps).

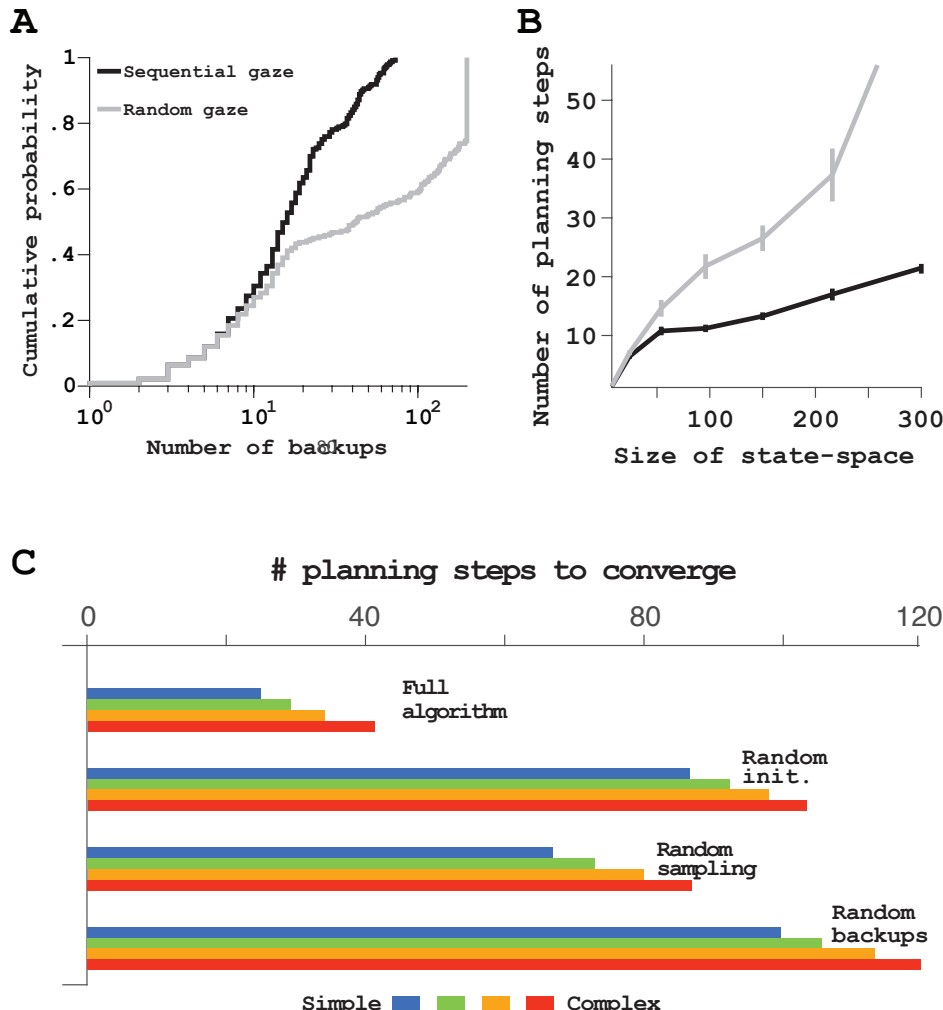

Figure 5: **Efficiency of the proposed algorithm. A**. Cumulative probability distribution of the number of backups across trials simulated using sequential information sampling (proposed algorithm, black) and using random information sampling (gray) strategies. Policy updates were performed sequentially in both cases. **B**. Mean number of backups as a function of the total number of states in the simulated mazes for the two algorithms shown in A. **C**. Number of planning steps of the proposed algorithm, compared against those obtained when ablating (by randomization) one feature of the algorithm (initialization, order of model updates, order of Bellman backups).

Techniques that use incremental backups such as prioritized sweeping are particularly useful for reducing the computational complexity of planning in large state spaces. Since the proposed algorithm does not use a prioritization scheme, we wanted to know whether it

could scale to large state spaces. To test this, we simulated trials by constructing mazes that differed in the total number of states ranging from 6 to 294. We found that the number of planning steps increased only marginally with the number of states (Figure 5B – black), and increase that was entirely accounted for by an increase in the average path length of trials in larger mazes. Moreover, the algorithm outperformed the variant with random sampling (Figure 5B – gray) suggesting that both sample efficiency and computationally efficiency are robust to the size of the state space. Finally, we probed whether the performance of the algorithm largely depended on appropriate initialization of the value function, ordering of model updates according the the value-gradient heuristic, or the sequence of Bellman backups by ablating each feature separately. We found that all three features were critical for efficient convergence (Figure 5C).

## 5. Limitations and future work

Although the proposed algorithm was successful at finding the optimal path in the environments we tested, it may yield suboptimal policies under certain conditions. Since active sensing is guided by local value gradients, one could construct mazes in which the initial direction of the gradient pushes the exploration towards a long-winding path. A rigorous theoretical treatment is required to understand the precise conditions under which this strategy is reasonable. Another direction we have not yet explored is a systematic model comparison against alternative algorithms such as prioritized sweeping or $n$-step look-ahead which may also co-exist with active sensing. Moreover, it would be intriguing to investigate how this algorithm is implemented neurally. Planning is associated with sequential neural activity in the hippocampus (Brown et al., 2016; Miller et al., 2017; Mattar and Daw, 2018; George et al., 2021; Zhu et al., 2023). Testing whether such neural dynamics could serve as a substrate for planning by active sensing is an important topic for future work. Finally, while the current study focused on visually-guided navigation, visual information might be either unnecessary or insufficient for some planning problems. Whether the proposed model can inspire useful computational strategies in such settings remains to be seen.

## 6. Conclusion

Planning and active sensing have traditionally been modeled separately. Planning has traditionally been characterized as a covert, internal information search over past experiences while active sensing is an overt, external search to gather new information. There is a growing realization that these two search strategies may be intertwined for temporally extended behaviors under naturalistic conditions (Lakshminarasimhan et al., 2020; Hunt et al., 2021). By analyzing gaze patterns, we find that human planning strategy during navigation is consistent with an algorithm in which both searches are coordinated and carried out simultaneously around the same set of states, a strategy we refer to as 'planning by active sensing'. The algorithm differs from traditional incremental approaches to dynamic programming in that planning (policy update) is performed in conjunction with learning (model update). It also differs from architectural frameworks such as 'Dyna' that allow for simultaneous learning and planning in that model updates are performed by visual sampling rather than physically interacting with the world which could prove too costly. Instead, the

strategy is conceptually similar to adaptive RTDP and identifies closely related real time algorithms as a promising direction for achieving human-like planning in machines.

## Acknowledgments

This work was supported by the NIH (1U19 NS118246, CRCNS 1R01 NS120407-01, 1R01 DC004260, 1R01 NS120407), the NSF NeuroNex Award (DBI-1707398) and the Gatsby Charitable Foundation (GAT3780).

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

# Appendix A. Appendix

## A.1. Experimental setup

Thirteen human participants participated in the experiments. All experimental procedures were approved by the Institutional Review Board at the university and all participants signed an informed consent form (IRB-FY2019-2599). Participants were seated on a swivel chair with 360° of freedom in physical rotation and navigated in a full-immersion hexagonal virtual maze with several obstacles. The stimulus was rendered at a frame rate of 90 Hz using the Unity game engine and was viewed through an HTC VIVE Pro virtual reality headset. The subjective vantage point was 1.72 m above ground level and the field of view of 110.1° of visual angle. Forward and backward translation was enabled via a continuous control CTI Electronics M20U9T-N82 joystick. Participants executed angular rotations inside the maze by turning their head, while the joystick input enabled translation in the direction in which the participant's head was facing. Obstacles and maze boundaries appeared as gray, rectangular slabs of concrete.

## A.2. Maze structure

The maze was a regular hexagon enclosing an area of approximately $260 m^2$ of navigable space. For ease of simulation and data analyses, the maze was imparted with a hidden triangular tessellation ('deltille') composed of $6n^2$ equilateral triangles, where $n$ determines the state space granularity. We chose $n = 5$, resulting in triangles with a side length of 2 meters, each of which constituted a state in the discrete state space. The maze contained several obstacles in the form of obstacles (0.4 m high) located along the edges between certain triangles (states). Outer boundary walls of height 2.5 m enclosed the maze. We chose five mazes (including an open maze) spanning a large range in average state closeness centrality, where closeness centrality is defined as the inverse average path length from state to every other state. On average, mazes with lower centrality will impose greater path lengths between two given states, making them more complex to navigate. The order of mazes presented to each participant was randomly permuted.

## A.3. Trial structure

At the beginning of each trial, a target in the form of a realistic banana appeared hovering 0.4 m over a state randomly drawn from a uniform distribution over all possible states. The joystick input was disabled until the participant foveated the target, but the participant was free to scan the environment by rotating in the swivel chair during the visual search period. About 200 ms after target foveation, the banana disappeared and participants were tasked with navigating to the remembered target location without time constraints. Participants were not given instructions on what strategy to use to complete the task. After reaching the target, participants pressed a button on the joystick to indicate that they have completed the trial. Feedback was displayed immediately after the button press.

## A.4. Reward structure

If participants stopped within the triangular state which contained the target, they were rewarded with two points. If they stopped in a state sharing a border with the target

state, they were rewarded with one point. After the participant's button press, the number of points earned on the current trial was displayed for one second at the center of the screen. The message displayed was 'You earned p points!'; the font color was blue if p=1 or p=2, and red if p=0. On skipped trials, the screen displayed 'You passed the trial' in red. In each experimental session, after familiarizing themselves with the movement controls by completing ten trials in a simplistic six-compartment maze (granularity $n = 1$), participants completed one block of 50 trials in each of five mazes. At the end of each block, a blue message stating 'You have completed all trials!' prompted them to prepare for the next block. Session durations were determined by the participant's speed and the length of the breaks that they needed from the virtual environment, ranging from 1.5–2 hr, sometimes spread across more than 1 day. Participants were paid 0.02/point for a maximum of 5 mazes x 50 trials/maze $\times$ 2 points/trial $\times$ 0.02/point $= 10$ dollars, in addition to a base pay of 10 /hr for their time (the average payment was 27.55 dollars).

## A.5. Sweep detection

Eye movements sweeps along the intended trajectory were detected by first calculating the point (x,y) on the trajectory closest to the location of gaze in each frame. For each trial, the fraction of the total trajectory length corresponding to each point was stored as a variable $f$, and periods when $f(t)$ consecutively ascended or descended were identified. For each period, we determined $m$, an integer whose magnitude denoted the sequence length and whose sign denoted the sequence direction ($+/$ for ascending/descending sequences). We then constructed a null distribution $P(m)$ describing the chance-level frequency of $m$ by selecting 20 random trials and recomputing $f$ based on the participant's trajectories on those trials. Sequential eye movements of length m where the CDF of $P(m)$ was less than $\alpha/2$ or greater than $1 - \alpha/2$ were classified as backward and forward sweeps, respectively. The significance threshold $\alpha$ was chosen to be 0.02. Compensating for noise in the gaze position, we applied a median filter of length 20 frames to both the true and shuffled $f$ functions. During post-processing, sweeps in the same direction that were separated by less than 25 frames were merged and sweeps were required to be at least 25 frames in length. To remove periods of fixation, the minimum variance in $f(t)$ values for all time points corresponding to the sweep was required to be 0.001. Finally, sweeps which did not cover at least 20% of the total trajectory length were removed from the analyses. This algorithm allowed for the automated detection of sequential eye movements pertaining to the prospective evaluation of trajectories which participants subsequently took.

## A.6. Quantification of sequence similarity

We quantified the similarity between the sequence of states *viewed* during planning and the sequence of states *visited* while subsequently navigating towards the goal using *ordering-based sequence similarity* (OSS) (Gomez-Alonso & Valls, 2008). Briefly, let $i = (x_{i,1}, ..., x_{i,card(i)})$ and $j = (x_{j,1}, ..., x_{j,card(j)})$ be two sequences of different lengths and let $S = (s_1, ..., s_n)$ be the set of all possible states comprising the elements of those sequences. The the (dis)similarity between two sequences $i$ and $j$ is quantified as:

$$d_{\text{OSS}}(i,j) = \frac{f(i,j) + g(i,j)}{\text{card}(i) + \text{card}(j)} \qquad (6)$$

where

$$g(i,j) = \text{card}(\{x_{ik}|x_{ik} \notin j\}) + \text{card}(\{x_{jk}|x_{jk} \notin i\}) \qquad (7)$$

measures the number of uncommon elements in the sequences and

$$f(i,j) = \frac{\sum_{k=1}^{n} |\sum_{p=1}^{\Delta} i_{(l_k)}(p) - j_{(l_k)}(p)|}{\max\{\text{card}(i), \text{card}(j)\}} \qquad (8)$$

measures the relative distance between the position of common elements across the two sequences. $d_{\text{OSS}}$ ranges between 0 (identical) and 1 (non-overlapping).

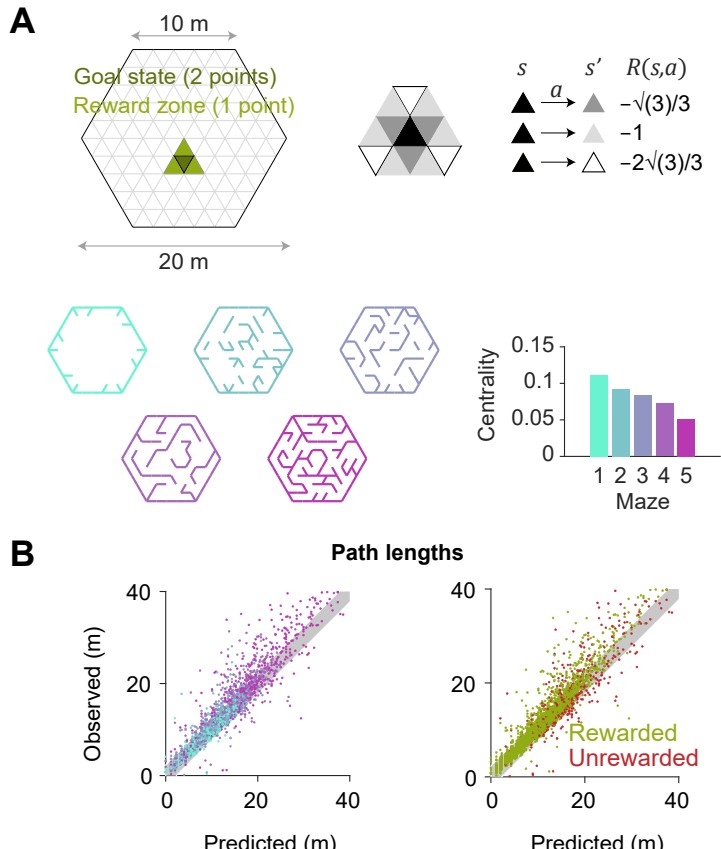

Figure 6: Participants exhibit near-optimal navigation performance across multiple environments. **A**. Top left: Mazes were regular hexagons of side length 10 m with a triangular tessellation of side length 2 m. Two points were rewarded if participants reached the goal state (green), and one point was rewarded if participants reached a state neighboring the goal state (light green). Top right: To incorporate twelve degrees of freedom in translation, value functions were computed using dynamic programming, whereby the cost of actions scaled in accordance with the center-to-center distance between states $s$ and $s'$ (pertaining to the transition which results from taking action $a$). Bottom left: Aerial view showing the layout of the mazes. Bottom right: Mazes ranged in mean state closeness centrality (mazes with higher complexity have lower centrality). **B**. Comparison of the empirical path length against the path length predicted by the optimal policy calculated using dynamic programming. The gray shaded region denotes the width of the outer reward zone (see panel A). Left: Data points are colored in accordance to the colors of each maze as depicted in A. Right: Unrewarded trials (red) vs rewarded trials (green) had similar path lengths. For both plots, all trials for all participants and all mazes are superimposed. Trials with observed path length lower than optimal correspond to paths that do not terminate exactly on the goal state, but some of these may terminate within the reward zone.

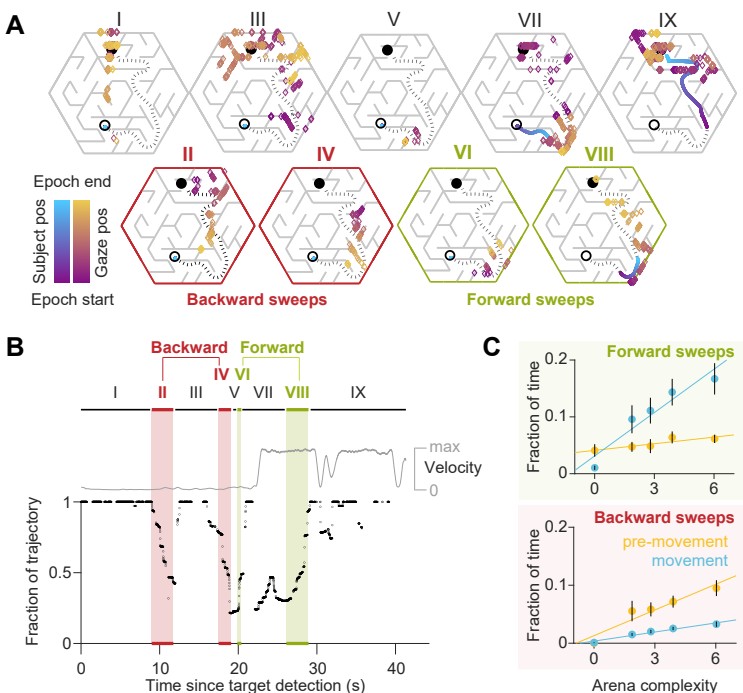

Figure 7: Gaze traveled forwards and backwards along the intended trajectory. **A**. Spatial locations of gaze positions (the arrow of relative time within each window increases from violet to orange) and participant positions (violet to blue) during individual time windows demarcated in panel B. Panels in the bottom row correspond to time periods corresponding to sweeps. The participant's trajectory from the starting location (open black circle) to the goal (closed black circle) is denoted by a black dashed line. **B** Time-series of the points on the trajectory that were closest to the participant's gaze on each frame, expressed as a fraction (0: start of trajectory, 1: end of trajectory) during one example trial. Only frames during which the gaze position fell within 2 m of the trajectory are plotted. The gray trace shows the movement velocity of the participant during this trial. Red and green shaded regions highlight time windows during which the sweep classification algorithm detected backward and forward sweeps, respectively. In this trial, there were two backward sweeps before movement, and one forward sweep each before and during movement. **C**. Across participants, the fraction of time spent sweeping in the forward and backward directions increased with the maze complexity. Error bars denote ±1 SEM. Figure 2F of the manuscript does not distinguish between forward and backward sweeps.

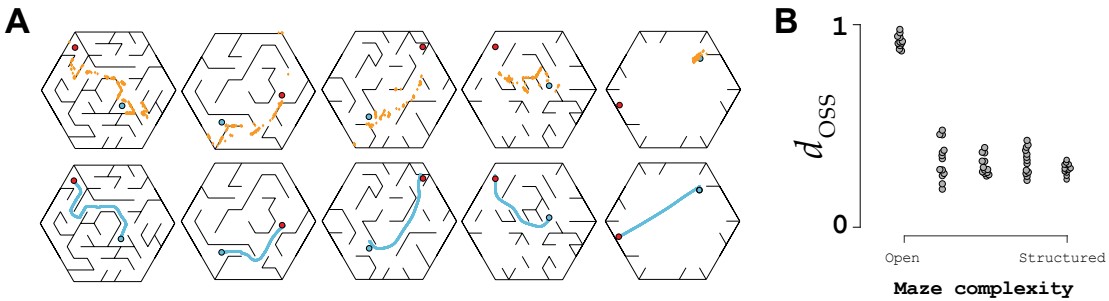

Figure 8: Similarity between active sensing strategy during planning and navigation. **A**. Example trials showing the spatial pattern of gaze during planning (orange) and path subsequently taken during navigation (blue) in different mazes. **B**. The ordering-based sequence similarity, $d_{\mathrm{OSS}}$ (similar sequences have lower values) between the states viewed during planning and states visited during navigation for individual participants ($n = 13$).

