# OpenReview forum: "Planning by Active Sensing"
_NeurIPS.cc/2023/Workshop/Gaze_Meets_ML — Gaze Meets ML 2023 Oral_

### Official Review · Reviewer_5DVw · 2023-10-21
**Impressive novel maze-planning algorithm inspired by human visual planning**

**Rating:** 9
**Confidence:** 4

**Review:**

This is an impressive paper that develops a novel planning algorithm for “planning by active sensing” in mazes. As opposed to the majority of the planning algorithm literature, this method uses efficient visual samples to learn a model of the environment before actually navigating to the target. This behavior is inspired by their study of human visual maze navigation in VR. Overall, the algorithm represents a much-needed fusion of planning and navigation with sensing (via eye movements). In addition to its technical rigor, thoroughness, and strong results, the paper is written well and clearly expresses its aims and conclusions.

There are several things I like about this paper:
* The authors do a very good job of contextualizing various aspects of the problem they aim to solve, from the perspective of prior works and specific current knowledge gaps.
* The final algorithm is shown to work well while being more computationally efficient than traditional approaches.
* The algorithm is a great example of taking inspiration from biological aspects of human vision; the authors ran an actual behavioral study and drew planning insights from that. It is a case of empirical rather than fuzzy, abstract inspiration, which, to my knowledge, is all too rare in neuro-inspired algorithm literature.

As a minor note, the term “backup” is first used in section 4.3 without explanation and is relatively important throughout the rest of the paper. Additionally, perhaps I missed this, but it would be good to explicitly clarify that one can only collect pre-navigation visual samples in this fashion if one can actually see the whole maze from a high vantage point, and what the implications of this requirement are on the applicability of this planning approach to other problems in planning.

---

### Official Review · Reviewer_g4WB · 2023-10-23
**Clear conceptual explanations, illustrations and compelling results**

**Rating:** 8
**Confidence:** 3

**Review:**

The paper has very clear treatment of salient aspects of active sensing. The clear explanations and illustration for **planning** algorithm was useful.
Some feedback to consider:
 - In how many instances did the greedy algorithm make a significantly suboptimal choice where they had to traceback a significant number of states especially in mazes with large number of space states?
 - During planning, how much does the experience play a role in optimal plan selection and subsequent policy update?
 - In the experiment, the vantage points differed across trials. Did higher vantage points trails result in better plan?
 - The authors had positive result that the algorithm performs computationally efficiently with large space states. Does the larger
   space states affect planning?

---

### Official Review · Reviewer_1iN9 · 2023-10-23
**Great paper - clear accept**

**Rating:** 10
**Confidence:** 3

**Review:**

Based on insights from a human study for maze navigation, an algorithm is proposed for intertwined planning and sensing.

No weaknesses, clear accept.

---

### Meta-Review · Area_Chair_dAxW · 2023-10-26

**Recommendation:** Accept (Oral)
**Confidence:** 5

**Metareview:**

The paper proposes efficient planning under uncertainty in complex environments achieved by human-like active sensing, as demonstrated in maze navigation studies, using a sample-efficient algorithm that dynamically updates a value map to iteratively choose eye movements for optimal path planning.

All reviewers and I agree this paper constitutes a very good contribution to the workshop. Please revise the paper accordingly to minor suggestions from reviewers.

---

### Decision · Program_Chairs · 2023-10-26

Accept (Oral)